# Sarcopenia after Roux-en-Y Gastric Bypass: Detection by Skeletal Muscle Mass Index vs. Bioelectrical Impedance Analysis

**DOI:** 10.3390/jcm11061468

**Published:** 2022-03-08

**Authors:** Georgi Vassilev, Christian Galata, Alida Finze, Christel Weiss, Mirko Otto, Christoph Reissfelder, Susanne Blank

**Affiliations:** 1Department of Surgery, Medical Faculty Mannheim, Heidelberg University, 68167 Mannheim, Germany; alida.finze@umm.de (A.F.); mirko.otto@umm.de (M.O.); christoph.reissfelder@umm.de (C.R.); susanne.blank@umm.de (S.B.); 2University Thoracic Center Mainz, Medical Faculty Mainz, 55131 Mainz, Germany; christian.galata@unimedizin-mainz.de; 3Department of Medical Statistics, Medical Faculty Mannheim, Heidelberg University, 68167 Mannheim, Germany; christel.weiss@medma.uni-heidelberg.de

**Keywords:** skeletal muscle mass, bioelectrical impedance analysis, Roux-Y gastric bypass

## Abstract

Background: In sarcopenic patients the skeletal muscle reduction is the primary symptom of age- or disease-related malnutrition, which is linked to postoperative morbidity and mortality. The skeletal muscle mass index (SMI) from magnet resonance imaging (MRI) is increasingly used as a prognostic factor in oncologic and surgical patients, but under-represented in the field of obesity surgery. The bioelectrical impedance analysis (BIA), on the other hand is a commonly used method for the estimation of the body composition of bariatric patients, but still believed to be inaccurate, because of patient-related and environmental factors. The aim of this study was to compare the postoperative SMI values as a direct, imaging measured indicator for muscle mass with the BIA results in patients undergoing Roux-en-Y gastric bypass (RYGB). Methods: We performed a prospective single-center trial. Patients undergoing RYGB between January 2010 and December 2011 at our institution were eligible for this study. MRI and BIA measurements were obtained 1 day before surgery and at 6, 12 and 24 weeks after surgery. Results: A total of 17 patients (four male, 13 female, average age of 41.9 years) were included. SMI values decreased significantly during the postoperative course (*p* < 0.001). Comparing preoperative and postoperative measurements at 24 weeks after surgery, increasing correlations of SMI values with body weight (r = 0.240 vs. r = 0.628), phase angle (r = 0.225 vs. r = 0.720) and body cell mass (BCM, r = 0.388 vs. r = 0.764) were observed. Conclusions: SMI decreases significantly after RYGB and is correlated to distinct parameters of body composition. These findings show the applicability of the SMI as direct imaging parameter for the measurement of the muscle mass in patients after RYGB, but also underline the important role of the BIA, as a precise tool for the estimation of patients’ body composition at low costs. BIA allows a good overview of patients’ status post bariatric surgery, including an estimation of sarcopenia.

## 1. Introduction

Obesity is a global health challenge and the main risk factor for diseases such as type 2 diabetes, cardiovascular morbidity, hypertension, sleep apnea, skeletal pain syndromes, psychological disorders, cancer and even early death [1]. Bariatric surgery has proven to be an effective strategy in treating obesity [2].

The main objective of Roux-en-Y gastric bypass (RYGB) is weight loss and improvement of metabolic comorbidities. Together with weight reduction, bariatric surgery leads to a change in body composition. Especially the fat mass decreases throughout the first months after surgery. Within this period, body cell mass (BCM), lean body mass (LBM), and absolute muscle mass and strength often also decrease [3,4,5,6,7]. The postoperative changes of those parameters are associated with weight loss, physical performance and risk of malnutrition and can be direct or indirect signs of a reduction in muscle mass [8,9].

Taking this into account, it is important to monitor the body composition and the skeletal muscle mass before and after bariatric surgery. There are different tools available to measure or estimate the BCM, LBM and the skeletal muscle status, such as bioelectrical impedance analysis (BIA), Dual-energy X-ray absorptiometry (DxA), handgrip dynamometry (HD) or imaging techniques including MRI and CT scan [10,11,12].

The analysis of single-layer images (CT scan or MRI) is used to quantify whole body muscle mass in vivo. The cross-sectional area of skeletal muscles (SMA, cm^2^) at the level of the third lumbar vertebra (L3), normalized for height, can be used to calculate the skeletal muscle index (SMI, cm^2^/m^2^), which is linearly related to the whole-body muscle mass [13,14].

BIA is commonly performed for the evaluation of pre- and postoperative body composition delivering the parameters BCM, extracellular mass (ECM), LBM and body fat. The phase angle reflects the quality of LBM [15]. The BIA provides accurate values comparable to those obtained by dual-energy X-ray absorptiometry (DXA) at low cost [10]. It measures body component resistance and capacitance by recording a voltage drop in applied current. Capacitance causes the current to lag behind the voltage, which creates a phase shift. This shift is quantified geometrically as the angular transformation—the phase angle [5].

The general loss of muscle mass is defined as sarcopenia. The term “sarcopenic obesity” describes the co-presence of sarcopenia and obesity. SMI is a surrogate parameter for sarcopenia and thus, a reduction of SMI is related to physical disability, increased morbidity and even mortality in surgical patients. This has been investigated mostly in geriatric and oncologic patients [16,17,18,19].

In patients after bariatric surgery, the role of SMI pre- and postoperatively is rarely described in literature. The correlation between SMI- and BIA- measurements remains controversial [20].

This study aims to investigate if the BIA as a common technique for estimating the body composition is still robust in comparison with the SMI measured by MRI in a cohort of patients undergoing RYGB.

## 2. Materials and Methods

***Patients*:** Between January 2010 and December 2011, an open, prospective, single center study was conducted at our institution investigating postoperative changes in body composition in bariatric patients via MRI and BIA measurements. Patients undergoing RYGB were included in the study. Further inclusion criteria were BMI 35–60 kg/m^2^, body weight < 200 kg, adequate patient compliance, waist circumference < 136 cm (MRI gantry diameter) and age > 18 years. Patients with contraindications for MRI or not willing or able to give informed consent were excluded from the study. The primary analysis of this study has been published previously [12]. For this post-hoc analysis, the SMI was measured retrospectively using the MRI studies performed in the prospective trial.

***Bioelectrical Impedance Analysis*** Bioelectrical impedance measurements were conducted according to standard protocols using a multiple frequency four-lead BIA instrument (Nutriguard-M, Data Input GmbH, Pöcking, Germany). Calculations for phase angel, body cell mass (BCM), extracellular mass (ECM), lean body mass (LBM), ECM/BCM, body fat (BF) and total body water (TBW) were made using the Nutriguard Plus software (version 5.4, Data Input GmbH, Pöcking, Germany).

***Magnetic Resonance Imaging*:** Abdominal MRI exams were obtained using a 1.5 Tesla whole-body scanner (MAGNETOM Avanto, Siemens Healthengineers, Erlangen, Germany) following standard clinical protocols. The anatomical coverage was from the upper edge of the liver to beneath the third lumbar vertebra level.

***Skeletal Muscle Mass Index*:** SMI was determined as published previously [20]. The SMI for each individual was calculated from MRI using two adjacent axial images within the same series. Total muscle cross-sectional area (cm^2^) at L3 was determined and averaged for each patient: The lumbar vertebrae 3 was identified, and the following muscles were selected using aycan workstation pro software (version 3.12.000, aycan Digitalsysteme GmbH, Würzburg, Germany): rectus abdominis, abdominal (lateral and oblique), psoas, and paraspinal (quadratus lumborum, erector spinae). Muscle area in centimeters squared (cm^2^) was calculated and then normalized for patient’s height in meters squared (m^2^) and reported as lumbar SMI (cm^2^/m^2^).

***Statistical analysis*:** Mean and standard deviation were calculated for quantitative variables. Qualitative variables were quoted as absolute numbers and relative frequencies. With the range or interquartile range, the median was presented for skewed or ordinally scaled parameters. Changes in parameters between measurements were examined using analysis of variance for repeated measurements. Post hoc analyses for pairwise mean comparisons were performed using the Scheffé method. For correlation analyses, Pearson correlation coefficient was determined. A test result was considered statistically significant if *p* < 0.05. Statistical analyses were performed using the SAS statistical analysis software (SAS release 9.4, Cary, NC, USA).

## 3. Results

A total of 17 patients were included in the study; four male and 13 female. The average age of the patients was 41.9 years. Mean initial body weight was 119.34 ± 11.86 kg and mean initial BMI was 42.96 ± 4.5 kg/m^2^. All patients underwent RYGB. Among other elements of the preoperative preparation like psychological, endocrinology- and nutrition expert assessment, every patient has documented at least 2.5 h of self-organized physical activity per week. When considering comorbidities, seven patients had no secondary disease, five had hypertension, four had sleep apnea, two had diabetes and one had GERD and knee arthrosis, respectively (Table 1).

There were no postoperative surgical complications. MRI, as well as BIA, was performed one day before surgery (t1) as well as 6 weeks (t2), 12 weeks (t3) and 24 weeks (t4) after surgery. Measurements at t1 and t2 were complete for all patients while at t3 and t4 they were only complete in 11 and 7 patients, respectively.

Table 2 shows the mean values of the respective parameters measured by BIA and the SMI measured by MRI as described above. In Table 3 the *p*-values for the respective comparisons are given. Changes in body weight and BMI are significant between t1 and t2, t2 and t3, but not between t3 and t4. Overall, most pronounced changes are observed between t1 and t2 (before surgery and 6 weeks after surgery). As expected, the body fat is significantly reduced after bariatric surgery. We did not find any further significant reduction between t3 and t4. Nevertheless, the LBM as well as BCM and ECM/BCM Index changed after surgery with a significant reduction of LBM and BCM between t1 and t2 and an almost significant reduction when comparing t2 to t4. The reduction of BCM results in an increase of the ECM/BCM Index, indicating malnutrition. The muscle mass also decreased over the observed time period being displayed by SMA measurement in BIA and SMI measurement in MRI imaging. The reduction of muscle mass is significant comparing the status before and after surgery but also between t2 and t4.

Figure 1, Figure 2 and Figure 3 reveal the quartiles, interquartile range (IQR) and outliers for the variables BMI, SMI and SMA for different time points.

Table 4 summarizes the Pearson Correlation Coefficient r for comparison of SMI with the parameters of body composition measured by BIA. No relevant correlation can be observed between BMI and SMI, but we found a correlation between the phase angle, BCM, ECM/BCM—Index and SMI. The higher the phase angle, the higher the SMI. The same applies to BCM. The higher the ratio of ECM/BCM, the lower the SMI.

Applying the cut-offs for sarcopenia introduced by Prado et al. [21] (SMI < 52.4 cm^2^/m^2^ for men and <38.5 cm^2^/m^2^ for women), 12% of the patients were sarcopenic before surgery (one man and one woman), 17% were sarcopenic at 6 weeks after surgery, 45% at 12 weeks after surgery and 57% at 24 weeks after surgery.

## 4. Discussion

In the current study, we investigated the changes in the SMI measured on a single L3- MRI layer as a direct indicator for the skeletal muscle mass of obese patients undergoing a RYGB procedure compared to BIA. To our knowledge, the direct comparison of those two methods is novel. The SMI is rarely discussed in literature, concerning bariatric surgical patients, but it is widely recognized as a direct parameter of the muscle mass status, because of the high accuracy and low susceptibility to external factors, in many other fields of medicine [21]. BIA on the other side is an often-used tool, which is still not considered sufficiently reliable, because of dependence on patient related and environmental factors, such as fasting and exercise status, previously to the measurement. Our results show a strong correlation between the SMI and the main parameters of the BIA (phase angle, LBM, BCM and the ECM/BCM—Index), which indicates that both methods are comparable in terms of estimating the change in body composition after bariatric surgery. These findings are in line with a publication of Walowski et al., considering that single computed tomography or MRI layers and appendicular lean soft tissue by DXA or BIA can be used as a valid substitute for total skeletal muscle mass. All diagnostics show a high correlation concerning body composition with results from whole body imaging in cross-sectional and longitudinal analyses [22]. BIA is a very feasible and inexpensive method for determination of the body composition. The determination of SMI by MRI is a very exact method in patients with mild obesity, but still MRI is more expensive and more time consuming than BIA. Our results clearly show that BIA, performed under standardized setting, has a good applicability and precision as a direct, imaging measured method as the SMI determination. Both methods, BIA and MRI, can be used for the estimation of body composition and presence of sarcopenia in patients after RYGB. The reliability of BIA has been previously described by our study group [5,11,12]. In our center we routinely use the BIA throughout the preparation of our patients for bariatric procedure, as well as in the follow up. This technique is feasible at low costs and the present study shows, that its results are resilient in comparison to the SMI derived from MRI. We are not doing MRI exams routinely in our patients, but we determine it in case of preexisting cross- sectional imaging.

Lee et al. also described that SMI values significantly correlate to BIA parameters among RYGB- patients, but not with the percent decrease after the procedure. These findings are in principle in line to our results, even though only one CT- scan after 6 months was performed postoperatively [20].

The reduction of SMI as well as BCM, LBM and phase angle in the first six months after RYGB, detected in our study, is in line with the findings of Alba et al. The authors also describe a significant decline of total LBM and absolute muscle strength, along with weight loss and fat mass reduction during the first year after RYGB [4]. Davidson et al. also demonstrated a decrease of SMI and fat free mass (FFM) during the phase of extensive weight loss in the first year after RYGB, but subsequent changes in MRI- measured muscle mass were minimal during the further follow up of 4 years [23]. According to these results, LBM and skeletal muscle mass reduction occurs frequently after bariatric surgery and mainly during the first year after surgery. In the meantime, Alba et al. described that even during the first year after RYGB, the decline of the muscle mass does not necessarily lead to poor clinical status of the patients. Their study showed a significant improvement in physical performance tasks despite a decrease of muscle mass. This fact could be explained by changes in biomechanics, which simply make it easier for a person to move around after weight loss. Nonetheless, maintaining more muscle mass or strength leads to greater functional improvements, and future research should address a range of strategies to optimize postoperative physical performance [4].

Two of our patients (11.8%) were sarcopenic according to the Prado- definition [24] before RYGB- procedure. Both patients were still sarcopenic 6 months after surgery. At that time point 57% of the examined patients were sarcopenic. Similar findings were made by a French group, detecting 32% obese patients with sarcopenia using SMI measured by MRI, one year after laparoscopic sleeve gastrectomy. However, only 8% of this cohort was in a sarcopenic condition before surgery [25].

The combination of low muscle mass and strength with obesity can further deteriorate the health status and physical performance of bariatric patients. Still, to date, the exact clinical meaning of these findings remains unclear. Sarcopenia seems to occur frequently in combination with obesity and is deteriorated in the early phase after bariatric surgery, indicating a special need for detection prior to surgery and an intense follow-up during the postoperative period. Structured programs, including an ongoing nutritional counseling and even structured rehabilitation programs, might be necessary to prevent patients from developing further sarcopenia and malnutrition. Hansen et al. demonstrated the important role of physical activity and exercise intervention in order to improve postoperative health benefits in terms of changes in body weight and fat mass, muscle mass and strength and physical fitness [26]. In contrast to our findings, Zamboni et al. reported that the “sarcopenic obesity” seems to play an important role in elderly patients, causing age- related gain of fat tissue and loss of muscle mass and also elderly subjects having a great health risk due to sarcopenic obesity [27]. In addition, such interventions lead to a better preservation of muscle strength, muscle mass, endurance capacity, and bone mineral density as well as greater quality of life [28]. Previous BIA studies clearly explain the importance of the preoperative determination of body composition and muscle mass status among bariatric surgical patients, describing the predictive value of the phase angle (parameter of the BIA) on postoperative body composition and potential weight loss [5,15].

## 5. Conclusions

Sarcopenia is a major problem in patients with obesity and can deteriorate further after bariatric surgery. Our data verify the accuracy of the BIA- parameters for muscle mass in comparison to the exact measurement of the SMI in single L3 layer of the abdomen. Both methods can detect the condition of sarcopenia in bariatric patients as an important factor for body composition before and after surgery. Patients should be screened for a reduction in muscle mass preoperatively as well as during long-term follow-up. Further, prospective trials are needed to investigate the exact clinical relevance of short-term and long-term sarcopenia after surgery.

## 6. Limitations

Our study has some limitations, one of them being the relatively small number of participants and the number of patients lost to follow-up during the end of the study. The small sample size of this study and the heterogeneity of our patient cohort in terms of gender, BMI and age did not allow us to perform more specific or complex statistical tests as multivariate regression analysis to reinforce our statement. Still, we were able to provide sequential BIA and SMI by MRI, which allowed us to give an overview of the development of body composition and muscle mass in the first months after RYGB.

## Figures and Tables

**Figure 1 jcm-11-01468-f001:**
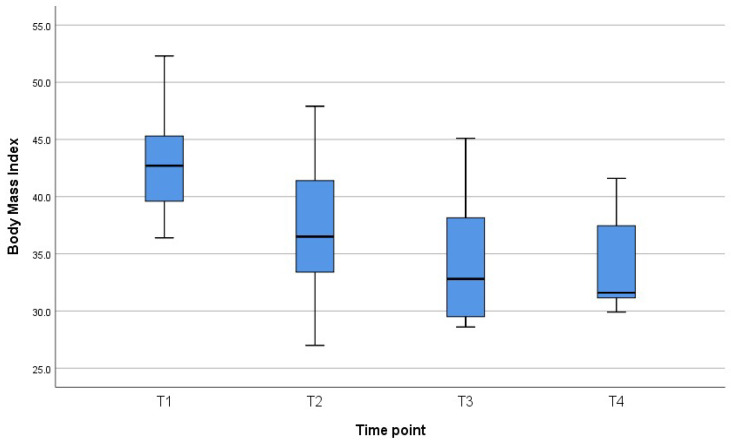
Box plot for BMI for different time points.

**Figure 2 jcm-11-01468-f002:**
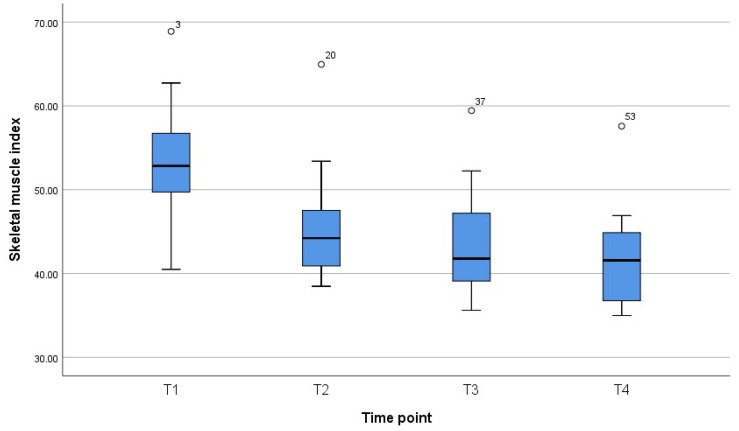
Box plot for SMI for different time points.

**Figure 3 jcm-11-01468-f003:**
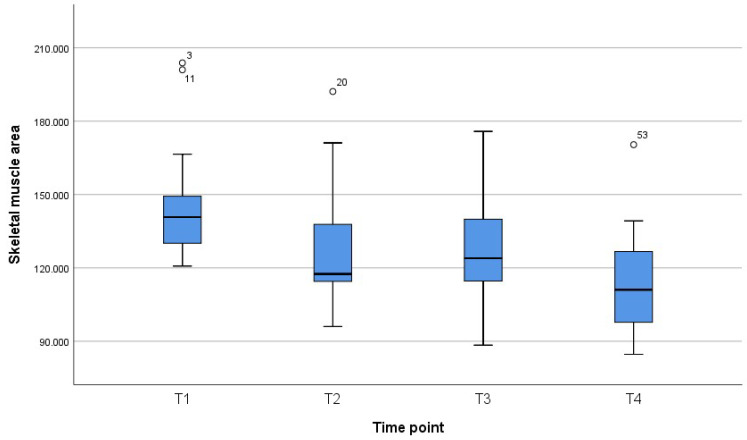
Box plot for SMA for different time points.

**Table 1 jcm-11-01468-t001:** Demographic characteristics of the respondents.

Demographic Characteristics	(*n* = 17)
Age	
Mean ± SD (Range (Max − Min))	41.9 ± 11.1 (35 (61 − 26))
Age group	
<=35	6 (35.3)
36–46	6 (35.3)
>=47	5 (29.4)
Gender	
Male	4 (23.5)
Female	13 (76.5)
Initial body weight (kg)	
Mean ± SD (Range (Max − Min))	119.34 ± 11.86 (47.6 (144.1 − 96.5))
Initial BMI (kg/m^2^)	
Mean ± SD (Range (Max − Min))	42.96 ± 4.5 (15.9 (52.3 − 36.4))
Initial SMI (cm^2^/m^2^)	
Mean ± SD (Range (Max − Min))	52.65 ± 7.06 (28.39 (68.89 − 40.5))
Comorbidities	
No Secondary disease	7 (50.0)
Hypertension	5 (35.7)
Sleep Apnea	4 (28.6)
Diabetes	2 (14.3)
GERD	1 (7.1)
Knee arthrosis	1 (7.1)

**Note:** The value is shown as mean ± sd (range) or *n* (%). **Abbreviation:** BMI, body mass index; SMI, skeletal muscle index; GERD.

**Table 2 jcm-11-01468-t002:** Body composition and skeletal muscle index at the different time points.

	t1	t2	t3	t4
Body weight (kg)	119.34 ± 11.86	103.67 ± 14.89	97.25 ± 10.87	92.59 ± 8.96
BMI (kg/m^2^)	42.96 ± 4.5	37.31 ± 5.69	34.72 ± 5.8	34.33 ± 4.62
Basal metabolic rate (kcal)	1685.29 ± 171.36	1558.24 ± 186.76	1546.36 ± 205.97	1547.14 ± 248.98
Phase angle (°)	6.38 ± 0.88	5.56 ± 0.93	5.31 ± 1.01	5.7 ± 1.26
TBW (kg)	44.39 ± 7.58	44.14 ± 7.64	44.57 ± 6.55	43.09 ± 7.15
LBM (kg)	63.38 ± 10.34	60.31 ± 47.30	60.89 ± 8.93	58.87 ± 9.79
ECM (kg)	29.55 ± 5.74	30.5 ± 5.87	31.52 ± 4.45	29.37 ± 4.81
BCM (kg)	33.83 ± 5.45	29.81 ± 5.90	29.38 ± 6.55	29.51 ± 7.83
Index (ECM/BCM)	0.88 ± 0.13	1.04 ± 0.19	1.11 ± 0.25	1.06 ± 0.36
BF (kg)	55.96 ± 6.97	43.36 ± 8.99	36.35 ± 7.79	33.71 ± 6.45
BF (%)	47.02 ± 5.04	41.70 ± 6.01	37.28 ± 6.20	36.59 ± 6.66
SMA (cm^2^)	146.73 ± 23.96	127.82 ± 24.71	124.22 ± 23.76	116.42 ± 29.37
SMI (cm^2^/m^2^)	52.65 ± 7.06	45.67 ± 6.62	43.84 ± 7.14	42.48 ± 7.86

Results are presented as mean ± standard deviation. t1 = before surgery, t2 = 6 weeks after surgery, t3 = 12 weeks after surgery, t4 = 24 weeks after surgery. BMI = body mass index, TBW = total body water, LBM = lean body mass, ECM = extracellular mass, BCM = body cell mass, BF = body fat, SMA = skeletal muscle area, SMI = skeletal muscle index.

**Table 3 jcm-11-01468-t003:** Comparison of BIA parameters between the different time points.

t	Body Weight	BMI	Basal Metabolic Rate (kcal)	Phase Angle	TBW	LBM	ECM	BCM	ECM/BCM	BF (kg)	BF (%)	SMA	SMI
**1 vs. 2**	<0.0001	<0.0001	<0.0001	0.0007	0.0002	0.0002	0.6115	<0.0001	0.0075	<0.0001	0.0002	<0.0001	<0.0001
**1 vs. 3**	<0.0001	<0.0001	<0.0001	0.0002	<0.0001	<0.0001	0.5693	<0.0001	0.0013	<0.0001	<0.0001	<0.0001	<0.0001
**1 vs. 4**	<0.0001	<0.0001	<0.0001	0.0052	<0.0001	<0.0001	0.9972	<0.0001	0.0079	<0.0001	<0.0001	<0.0001	<0.0001
**2 vs. 3**	0.0032	0.0045	0.4868	0.7336	0.7074	0.7029	0.9939	0.452	0.6751	0.0038	0.0074	0.5178	0.5735
**2 vs. 4**	<0.0001	<0.0001	0.0569	0.9557	0.0054	0.005	0.7076	0.0636	0.7784	0.0002	0.0015	0.0298	0.0416
**3 vs. 4**	0.1042	0.076	0.5251	0.9863	0.0658	0.0626	0.6024	0.5838	1	0.3965	0.7147	0.3509	0.3857

*p*-values for comparison between the respective time points. t1 = before surgery, t2 = 6 weeks after surgery, t3 = 12 weeks after surgery, t4 = 24 weeks after surgery. BMI = body mass index, TBW = total body water, LBM = lean body mass, ECM = extracellular mass, BCM = body cell mass, BF = body fat, SMA = skeletal muscle area, SMI = skeletal muscle index.

**Table 4 jcm-11-01468-t004:** Correlation of SMI with BIA parameters.

t	Body Weight	BMI	Basal Metabolic Rate (kcal)	Phase Angle	TBW	LBM	ECM	BCM	ECM/BCM	BF (kg)	BF (%)	SMA
**1**	0.24085	0.38667	0.38526	0.22527	0.28819	0.28748	0.15098	0.3879	−0.24203	−0.0167	−0.18213	0.74816
**2**	0.42458	0.30951	0.66135	0.51569	0.476	0.4753	0.18051	0.66573	−0.50681	0.1514	−0.14671	0.82661
**3**	0.27591	0.2205	0.65462	0.72809	0.40136	0.40183	−0.16068	0.66083	−0.71336	−0.07564	−0.24256	0.79288
**4**	0.62821	0.18605	0.76101	0.71963	0.58433	0.58561	−0.05668	0.76404	−0.64093	−0.01619	−0.30592	0.87446

Pearson Correlation Coefficient r. t1 = before surgery, t2 = 6 weeks after surgery, t3 = 12 weeks after surgery, t4 = 24 weeks after surgery. BMI = body mass index, TBW = total body water, LBM = lean body mass, ECM = extracellular mass, BCM = body cell mass, BF = body fat, SMA = skeletal muscle area, SMI = skeletal muscle index.

## Data Availability

The data presented in this study are available on request from the corresponding author. The data are not publicly available due tot he rules of the Medical Ethics Commission of our institution.

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
