# Peer review of "Sarcopenia after Roux-en-Y Gastric Bypass: Detection by Skeletal Muscle Mass Index vs. Bioelectrical Impedance Analysis"

_jcm, 2022, doi:10.3390/jcm11061468_

Round 1

Reviewer 1 Report

Dear authors,

please can you improve some wordings as:
page 3, line 120: of which 4 male and 13 female (correct writing, verb is missing)
page 3, line 125: followed by 5 have hypertension (correct writing and time!)

Moreover add more imformation or improve:
page 1, Introduction, line 38: disease - please describe in more detail the possible diseases!
page 2, line 80: study period 01/2010 - 12/2012: Patient recruitment is still 10 years ago, why are you publishing the data now? Is there any plausible cause? Don`t you follow up the strategy by BIA measurement in bariatric patients? Why not? What is about the Sleeve Technique, which is used very common in bariatric surgery? 
page 3, line 123: elements - about what elements are you writing?
page 4, lines 147-148: after 12 and 24 weeks - it makes understanding difficult, if you change between t1 and t2 as measuring points to 12 and 24 weeks as time intervals - please be consistent!

The reduced number of patients with wide differences in comorbidities and the evaluation a decade ago are the main limitations and therefore the value of your results really must be considered as questionable / inconclusive. Nevertheless it`s unequivocally that physical activity is most important for patients after bariatric surgery.

Kind regards!

Author Response

Comments to Reviewer #1:

Please can you improve some wordings as:
Page 3, line 120: of which 4 male and 13 female (correct writing, verb is missing) -> this sequence has been corrected
Page 3, line 125: followed by 5 have hypertension (correct writing and time!) -> this sequence has been corrected

Moreover add more information or improve:
Page 1, Introduction, line 38: disease - please describe in more detail the possible diseases! -> a more specific designation of the disease term has been added

Page 2, line 80: study period 01/2010 - 12/2012: Patient recruitment is still 10 years ago,

Why are you publishing the data now? Is there any plausible cause?

-> the raw data were collected at this early time point (10 y ago) and two studies with a focus on the estimation of fatty liver disease and the measurement of visceral fat in MR Imaging were published.

(Otto M, Farber J, Haneder S, Michaely H, Kienle P, Hasenberg T: Postoperative changes in body composition--comparison of bioelectrical impedance analysis and magnetic resonance imaging in bariatric patients. Obes Surg 2015, 25(2):302-309.  

Hedderich D, Hasenberg T, Haneder S, Schoenberg S, Kuecuekoglu Ö, Canbay A, Otto MEffects of Bariatric Surgery on Non-alcoholic Fatty Liver Disease: Magnetic Resonance Imaging Is an Effective, Non-invasive Method to Evaluate Changes in the Liver Fat Fraction. Obes Surgery, 2017 Jul; 27(7):1755-1762.)

At this timepoint the SMI was not in the focus of interest in bariatric medicine, so we did not evaluate it primarily. After the SMI has been proven to be a relevant prognostic marker in oncologic patients as well as patients with inflammatory bowel disease and an own publication about the role of SMI in Crohn’s disease we decided to rescreen our MRIs with the focus on the SMI in this group of bariatric patients (Galata C, Hodapp J, Weiss C, Karampinis I, Vassilev G, Reissfelder C, Otto M: Skeletal Muscle Mass Index Predicts Postoperative Complications in Intestinal Surgery for Crohn's Disease. JPEN J Parenter Enteral Nutr 2020, 44(4): 714-721.)

Don`t you follow up the strategy by BIA measurement in bariatric patients? Why not? -> In our center we routinely use the BIA throughout the preparation of our patients for bariatric procedure, as well as in the follow up. This technique is feasible at low costs and the present study shows, that its results are resilient in comparison to the SMI derived from MRI. We are not doing MRI exams routinely in our patients, but we determine it in case of preexisting cross- sectional imaging.   -> We now added this paragraph to the manuscript, because it is an important link to the practical implementation of our findings in the daily routine.

What is about the Sleeve Technique, which is used very common in bariatric surgery? -> Today the Sleeve Gastrectomy is a frequent procedure globally and also in our center. 10 years ago we did more Roux- y- gastric bypass, the second reason for including only patients with Roux-Y Gastric Bypass was to generate a homogeneous group of patients,. 

Page 3, line 123: elements - about what elements are you writing? -> Evaluation by the endocrinologist, psychology and nutrition-counseling professionals. A specification has been added to the manuscript.

Page 4, lines 147-148: after 12 and 24 weeks - it makes understanding difficult, if you change between t1 and t2 as measuring points to 12 and 24 weeks as time intervals - please be consistent! -> this sequence has been changed in the manuscript.

The reduced number of patients with wide differences in comorbidities and the evaluation a decade ago are the main limitations and therefore the value of your results really must be considered as questionable / inconclusive. Nevertheless it`s unequivocally that physical activity is most important for patients after bariatric surgery. -> We agree with the reviewer that the differences in comorbidities and the time point of the data collection are a weaknesses of our work, but the presentation of exactly measured SMI by MR- Images and reliable and consecutive measured BIA parameters, legitimate our study to be a part of the growing evidence about the different methods, needed to display the status of the lean body mass / muscle mass in patients, being treated for morbid obesity.

Reviewer 2 Report

The present study uses data from a previously published cohort in order to compare BIA and skeletal muscle mass index obtained by MRI in people undergoing bariatric surgery for morbid obesity. The authors have already published results regarding the MRI and BIA results. Thus, this paper only adds the element of SMI, which is only available for a fraction of the population, since there was very large attrition. Minor comment: references are not properly numbered.

Author Response

Comments to Reviewer #2:

The present study uses data from a previously published cohort in order to compare BIA and skeletal muscle mass index obtained by MRI in people undergoing bariatric surgery for morbid obesity. The authors have already published results regarding the MRI and BIA results. Thus, this paper only adds the element of SMI, which is only available for a fraction of the population, since there was very large attrition. Minor comment: references are not properly numbered. -> We agree with the reviewer that the SMI is not available for every single patient, but the positive correlation between this expensive method and the easier accessible BIA at low costs, increases the relevance of our study and makes it an important part of the evidence in this field.

-> We corrected the references.

Reviewer 3 Report

The article deals with important clinical and scientific issues and
is written in a comprehensible language.
The authors are able to discuss the obtained results.
I believe that the discussion is written correctly.
The description of the results is clearly written.
Well described groups.

Author Response

Comments to Reviewer #3:

The article deals with important clinical and scientific issues and is written in a comprehensible language. 
The authors are able to discuss the obtained results. 
I believe that the discussion is written correctly. 
The description of the results is clearly written. 
Well-described groups. 

-> We appreciate the comments of the reviewer.

This manuscript is a resubmission of an earlier submission. The following is a list of the peer review reports and author responses from that submission.

Round 1

Reviewer 1 Report

Dear authors!

The study is focussing on the analysis of changes in body composition mainly the skeletal muscle mass before and after bariatric surgery in 17 morbid obese patients without any data on age, obesity related comorbidities, visceral fat / waist circumference (waist/hip ratio), upper limb diameter etc. The data are collected approximately 10 years ago.

The manuscript is well prepared and clearly structured. The importance of the skeletal muscle mass in relation to morbidity and mortality is clearly described.

Statistical analyses of body composition estimated by MRI and BIA including the phase angle are described in detail and could clearly show the changes between 4 time points. Most important are the reductions of weight, BMI, basal metabolic rate, PhA, TBW, LBM, BCM, BF, SMA and SMI between T1 (before bariatric surgery) and T2 (6 weeks after bariatric surgery), T3 (12 weeks after bariatric surgery) as well as T4 (24 weeks after bariatric surgery) and the accordance between MRI - SMI related to BIA - SMA.

The aim of the study was to evaluate the skeletal muscle mass in morbid obese patients before and after bariatric surgery related to weight loss and to compare two different measuring methods, MRI - SMI and BIA - SMA.

Additional factors influencing the skeletal muscle mass before and after bariatric surgery including malnutrition, immobility/training program, surgical complications, infections, wound healing ... are not mentioned. This means there is no information on life style and disease / surgical related problems.  

Lines 78-79 describe sarcopenia and the term "sarcopenia obesity". It would be important to consider sarcopenia in relation to older age, immobility, malnutrition, inflammation, organ failure ... For example age relation is described in literature following: Ageing is associated with significant changes in body composition with a substantial reduction in fat-free mass and muscle mass and an increase in visceral fat, even if the body weight remains unchanged. (Zamboni, M, Mazzali, G, Fantin, F et al. (2008) Sarcopenic obesity: a new category of obesity in the elderly. Nutr Metab Cardiovasc Dis 18, 388–395.).
Therefore factors causing sarcopenia should be described and discussed also in connection with bariatric surgery in obese patients.

Lines 82 - 83 state that SMI is not well defined pre- and postoperatively. Recently a meta-analysis on different publications focussing on skeletal muscle mass in obese patients with or without bariatric surgery and a publication on fat mass reduction and skeletal muscle decrease measured by CT - SMI and BIA - SMA present clear data on body composition in obesity before and after bariatric surgery.
See literature:
1.) doi: 10.1007/s11695-021-05569-6. Comparison of Bioelectrical Impedance Analysis and Computed Tomography on Body Composition changes including Visceral  Fat after Bariatric Surgery in Asian Patients with Obesity. Lee JK et al Obes. Surg. 2021, 32 (10): 4243-50 and the conclusion: The SMI values showed significant correlations before and after surgery, but not with the percent decrease.
2.) doi: 10.1016/j.clnu.2021.07.035. A systematic review by Vincenzo O, Marra M, Sacco AM, Pasanisi F, Scalfi L.Clin Nutr. 2021 Sep;40(9):5238-5248.

Line 170 is related to the innovation of the manuscript, based on the  direct correlation of the two measuring methods, MRI and BIA, in determining the skeletal muscle mass. As mentioned above the recently published study on measuring skeletal muscle mass by CT and BIA maybe focus on a similar strategy (doi: 10.1007/s11695-021-05569-6. Comparison of Bioelectrical Impedance Analysis and Computed Tomography on Body Composition changes including Visceral  Fat after Bariatric Surgery in Asian Patients with Obesity. Lee JK et al Obes. Surg. 2021, 32 (10): 4243-50). This should be taken into account.

The above mentioned meta-analysis on Bioelectrical impedance (BIA) -derived phase angle in adults with obesity (A systematic review by Vincenzo O, Marra M, Sacco AM, Pasanisi F, Scalfi L.Clin Nutr. 2021 Sep; 40(9):5238-5248. doi: 10.1016/j.clnu.2021.07.035.) gives a good overview to the scientific work on skeletal muscle mass with focus on PhA in obese patients with and without comorbidities, without or after bariatric surgery and 2 publications presenting the success of training programs to prevent skeletal muscle decrease.  Are there any considerations in relation to the presented results in your manuscript?

Moreover the study by Teixeira J., Marroni C.A., Zubiaurre P.R., Henz A., Faina L., Pinheiro L.K. et al. Phase angle and non-alcoholic fatty liver disease before and after bariatric surgery. WJH. Nov 27; 12: 1004-1019 could show that the decline of PhA after RYGB was negatively associated with weight loss and positively with the decrease of BMI, skeletal muscle mass, FM, and visceral fat area. Could you make a comment on this.

In Conclusion, lines 234 - 243, you describe that both measuring methods can detect the condition of sarcopenia in bariatric patients before and after surgery. That can be confirmed by literature and your data are quiet well matching. The description on sarcopenia as major problem in obesity and the deterioration after bariatric surgery must be considered in more details as mentioned above, because not only obesity, but also a wide spectrum of different factors like older age, malnutrition, immobility, inflammation, organ failure, malignant diseases .... may effect the outcome. Therefore obese patients in younger age are not significantly effected by skeletal muscle decrease, but with age and appearing comorbidities "sarcopenic obesity" occurs more and more.

The manuscript should give more details on the obese patients, what should be done easily for the small number described in the manuscript. The recent publications on skeletal muscle mass in obese patients should be taken in consideration. Because data collection was approximately 10 years ago, changes in the guidelines for pre- und postsurgical programs and surgical techniques must be taken in consideration. Please could you comment on this.

See Introduction: lines 41 - 49 are describing the author instructions for the publication, please delete the lines!

Kind regards!

Author Response

Reviewer(s)' Comments to Author:

Reviewer #1:

The study is focusing on the analysis of changes in body composition mainly the skeletal muscle mass before and after bariatric surgery in 17 morbid obese patients without any data on age, obesity related comorbidities, visceral fat / waist circumference (waist/hip ratio), upper limb diameter etc. The data are collected approximately 10 years ago.

Line 129- 130 Average age and the comorbidities of the patients were added.

We do not have data about waist/ hip circumference or limb diameter in this group of patients.

Additional factors influencing the skeletal muscle mass before and after bariatric surgery including malnutrition, immobility/training program, surgical complications, infections, wound healing ... are not mentioned. This means there is no information on life style and disease / surgical related problems.

Line 134- 136 Information about the absence of surgical complications and the documented physical activity- program before the procedure was added.  

Lines 78-79 describe sarcopenia and the term "sarcopenia obesity". It would be important to consider sarcopenia in relation to older age, immobility, malnutrition, inflammation, organ failure ... For example age relation is described in literature following: Ageing is associated with significant changes in body composition with a substantial reduction in fat-free mass and muscle mass and an increase in visceral fat, even if the body weight remains unchanged. (Zamboni, M, Mazzali, G, Fantin, F et al. (2008) Sarcopenic obesity: a new category of obesity in the elderly. Nutr Metab Cardiovasc Dis 18, 388–395.).
Therefore factors causing sarcopenia should be described and discussed also in connection with bariatric surgery in obese patients.

Line 81- 84 A paragraph about the importance of “sarcopenic obesity” has been added. Zamboni et al. Citation.

Lines 82 - 83 state that SMI is not well defined pre- and postoperatively. Recently a meta-analysis on different publications focussing on skeletal muscle mass in obese patients with or without bariatric surgery and a publication on fat mass reduction and skeletal muscle decrease measured by CT - SMI and BIA - SMA present clear data on body composition in obesity before and after bariatric surgery.
See literature: 
1.) doi: 10.1007/s11695-021-05569-6. Comparison of Bioelectrical Impedance Analysis and Computed Tomography on Body Composition changes including Visceral  Fat after Bariatric Surgery in Asian Patients with Obesity. Lee JK et al Obes. Surg. 2021, 32 (10): 4243-50 and the conclusion: The SMI values showed significant correlations before and after surgery, but not with the percent decrease.
2.) doi: 10.1016/j.clnu.2021.07.035. A systematic review by Vincenzo O, Marra M, Sacco AM, Pasanisi F, Scalfi L.Clin Nutr. 2021 Sep;40(9):5238-5248.

The statement in line 82- 83 was corrected.

  • The first study is very interesting and hits exactly the point of our publication. For this reason we use it also for the discussion part. (Original Line 193 + in the discussion part

  • The second work, mentioned above focuses very on the role of the Phase angle in the body composition of obese, non- surgical patients. This is a very important and interesting study, but it has no direct relation to our main focus “SMI as a correlating factor to BIA”. Two studies of our research group (Vassilev G, Hasenberg T, Krammer J, Kienle P, Ronellenfitsch U, Otto M. The phase Angle of the bioelectrical impedance analysis as predictor of post- bariatric weight loss outcome. Obes Surg 2017 Mar;27(3): / [37]  Gerken ALH, Rohr-Kra€utle K-K, Weiss C, Seyfried S, Reissfelder C, Vassilev G, et al. Handgrip strength and phase Angle predict outcome after bariatric surgery. Obes Surg 2021;31(1):200e6. ) are used as citation in this paper.

Line 170 is related to the innovation of the manuscript, based on the  direct correlation of the two measuring methods, MRI and BIA, in determining the skeletal muscle mass. As mentioned above the recently published study on measuring skeletal muscle mass by CT and BIA maybe focus on a similar strategy (doi: 10.1007/s11695-021-05569-6. Comparison of Bioelectrical Impedance Analysis and Computed Tomography on Body Composition changes including Visceral  Fat after Bariatric Surgery in Asian Patients with Obesity. Lee JK et al Obes. Surg. 2021, 32 (10): 4243-50). This should be taken into account.

The above mentioned meta-analysis on Bioelectrical impedance (BIA) -derived phase angle in adults with obesity (A systematic review by Vincenzo O, Marra M, Sacco AM, Pasanisi F, Scalfi L.Clin Nutr. 2021 Sep; 40(9):5238-5248. doi: 10.1016/j.clnu.2021.07.035.) gives a good overview to the scientific work on skeletal muscle mass with focus on PhA in obese patients with and without comorbidities, without or after bariatric surgery and 2 publications presenting the success of training programs to prevent skeletal muscle decrease.  Are there any considerations in relation to the presented results in your manuscript?

As already mentioned the study of Lee et al was taken into consideration and is now part of our discussion section.

As already described the study of Vincenzo et al. does not include any correlation to SMI from MRI or CT – Scan measurements and is not directly connected to our focus. It is true that Phase angle from the BIA has some important predictive value in terms of weight loss or quality of muscle mass after surgery, but we do not know how it correlates with the SMI. This was not our goal in this paper. We already attend some works to the role of PhA as above mentioned and think that it will make the present study less understandable, when we try to mix this aspects into it.

Moreover the study by Teixeira J., Marroni C.A., Zubiaurre P.R., Henz A., Faina L., Pinheiro L.K. et al. Phase angle and non-alcoholic fatty liver disease before and after bariatric surgery. WJH. Nov 27; 12: 1004-1019 could show that the decline of PhA after RYGB was negatively associated with weight loss and positively with the decrease of BMI, skeletal muscle mass, FM, and visceral fat area. Could you make a comment on this.

The study of Teixeira et al is also very important contribution to the role of PhA around baritric surgical procedures. But it also will make our present paper less understandable and clear to the reader. We did not put PhA in focus in this work on purpose.

In Conclusion, lines 234 - 243, you describe that both measuring methods can detect the condition of sarcopenia in bariatric patients before and after surgery. That can be confirmed by literature and your data are quiet well matching. The description on sarcopenia as major problem in obesity and the deterioration after bariatric surgery must be considered in more details as mentioned above, because not only obesity, but also a wide spectrum of different factors like older age, malnutrition, immobility, inflammation, organ failure, malignant diseases .... may effect the outcome. Therefore obese patients in younger age are not significantly effected by skeletal muscle decrease, but with age and appearing comorbidities "sarcopenic obesity" occurs more and more.

We absolutely agree with your point. Sarcopenia itself seems to have a different consequence in different age and conditions (as above mentioned) of the patients, but the design of our study does not allow us to make a clear statement on all this factors. From original Line 223 on, we added a part about the age of the patients, even our subgroup is relatively young. We do not have patients over 65 y and more. But we think this is a very good and important point and has to be discussed in the Sarcopenia- Part. We do not have any patients with immobility, inflammation, organ failure etc. in our cohort, so we cannot ad any comments on this. Our cohort of patients does not include elderly patients- for this reason we do not make any statement on this very important point, in our conclusion.

The manuscript should give more details on the obese patients, what should be done easily for the small number described in the manuscript. The recent publications on skeletal muscle mass in obese patients should be taken in consideration. Because data collection was approximately 10 years ago, changes in the guidelines for pre- und postsurgical programs and surgical techniques must be taken in consideration. Please could you comment on this.

Patient information has been added. The recommended, new paper has been included. The pre-and postoperative programs for the patients are pretty similar to this early days- the number of patients and the level of professionalism has changed. All those 17 patients have had nutrition counseling (standard), endocrinology, psychology assessment and the self organized physical activity of at least 2.5 h / week (documented). The surgical technique for RYGB is very similar to the current, but the training level of our surgical team, respectively the duration of the procedures is extremely decreased.

Reviewer 2 Report

The publication was designed on the basis of a single center study, with a properly conducted statistical analysis. Unfortunately, the groups are very small, so the results achieved and the conclusions based on them are certainly and signifficantly biased. 

Author Response

Reviewer #2:

The publication was designed on the basis of a single center study, with a properly conducted statistical analysis. Unfortunately, the groups are very small, so the results achieved and the conclusions based on them are certainly and significantly biased. 

The second reviewer does not give us any specific suggestions, so there are unfortunately no exact answers we can provide.

We totally disagree with the statement of “certainly and significant bias”! We clearly point out the limitations of this work, but as you can read in the present study, there are only few paper with better or prospective design, to clear this very important field in bariatric surgery.

Still we agree with the need of a professional language editing of our article and will provide this together with the mdpi- team.

Round 2

Reviewer 2 Report

The study has still limitation and selected bias. At least the number of patients to overcome this bias should be more than thirty

At least sample size should be more than thirty to reduce selection bias